# Adaptations of the Genus *Bradyrhizobium* to Selected Elements, Heavy Metals and Pesticides Present in the Soil Environment

**DOI:** 10.3390/cimb47030205

**Published:** 2025-03-18

**Authors:** Joanna Banasiewicz, Aleksandra Gumowska, Agata Hołubek, Sławomir Orzechowski

**Affiliations:** 1Department of Biochemistry and Microbiology, Institute of Biology, Warsaw University of Life Sciences (SGGW), Nowoursynowska 159, 02-776 Warsaw, Poland; 2Faculty of Biology and Biotechnology, Warsaw University of Life Sciences (SGGW), Nowoursynowska 159, 02-776 Warsaw, Poland; s210051@sggw.edu.pl (A.G.); s210052@sggw.edu.pl (A.H.)

**Keywords:** biofertilizer, *Bradyrhizobium*, element, fabaceae, legume, heavy metal, pesticide, radionuclide, Rhizobium

## Abstract

Rhizobial bacteria perform a number of extremely important functions in the soil environment. In addition to fixing molecular nitrogen and transforming it into a form available to plants, they participate in the circulation of elements and the decomposition of complex compounds present in the soil, sometimes toxic to other organisms. This review article describes the molecular mechanisms occurring in the most diverse group of rhizobia, the genus *Bradyrhizobium*, allowing these bacteria to adapt to selected substances found in the soil. Firstly, the adaptation of bradyrhizobia to low and high concentrations of elements such as iron, phosphorus, sulfur, calcium and manganese was shown. Secondly, the processes activated in their cells in the presence of heavy metals such as lead, mercury and arsenic, as well as radionuclides, were described. Additionally, due to the potential use of *Bradyrhziobium* as biofertilizers, their response to pesticides commonly used in agriculture, such as glyphosate, sulfentrazone, chlorophenoxy herbicides, flumioxazine, imidazolinone, atrazine, and insecticides and fungicides, was also discussed. The paper shows the great genetic diversity of bradyrhizobia in terms of adapting to variable environmental conditions present in the soil.

## 1. Introduction

Rhizobia is a collective term for a group of soil bacteria that are best known for their ability to engage in mutualistic interactions with legume plants. They are diazotrophs that fix the atmospheric N_2_ gas into ammonia, constituting one of the most important sources of bioavailable nitrogen on Earth [1]. It is estimated that rhizobial bacteria in leguminous plants provide 70 to 80% of the plant’s nitrogen needs [2]. Additionally, rhizobia not only promote plant growth through nitrogen fixation in legumes, but also by various other mechanisms such as phytohormone and vitamin production, precipitated phosphorus solubilization, siderophore production, and by reduction in ethylene levels to cope with drought stress [3,4]. Furthermore, rhizobia can contribute to crop growth through interactions with other beneficial microorganisms and reduction in or prevention of harmful effects caused by pathogenic microorganisms, mainly through the synthesis of antibiotics or fungicidal compounds, and induction of systemic resistance to pathogens [4]. All these factors make rhizobia one of the best choices when it comes to using them as biofertilizers, which are an alternative to chemical fertilizers that have harmful effects on the environment [2]. Low-cost microbial inoculants are already known and have been successfully used to help improve plant growth, enhance their adaptation to soils contaminated with heavy metals, and control phytopathogens to enhance sustainable agriculture [5,6]. However, the bacterial strains used in vaccinations may not dominate during nodule formation due to competition with indigenous strains in the crop field, which are better adapted to the physicochemical conditions present there [6].

The most abundant and diverse genus of the rhizobia group is *Bradyrhizobium*. The genus represents a large and diverse assemblage of cosmopolitan bacteria capable of nodulating a variety of legumes from around the world and has been regarded as a model of legume–rhizobia symbiosis and an ecologically significant group [7,8]. Currently, more than 60 species of *Bradyrhizobium* are known [7]. Bradyrhizobia have biologically significant functions in soils, including nitrogen fixation, photosynthesis, denitrification, degradation of aromatic compounds, and many others [8,9,10,11,12,13,14]. Understanding the mechanisms of bradyrhizobial adaptation to independent existence in diverse and variable environments is becoming increasingly important and thus may reveal the genetic potential of this globally significant group of rhizobia [8]. It seems particularly important to know the influence of various substances present in soils on bradyrhizobial strains and the mechanisms that influence how this group of bacteria copes with them. The presented review describes the influence of selected elements, heavy metals and pesticides on bacteria from the *Bradyrhizobium* genus, as well as the molecular pathways present in the cells of these microorganisms, which are activated in response to the action of the above-mentioned factors.

## 2. The Influence of Selected Elements Present in Soil on Bradyrhizobia and Their Symbiotic Processes

As mentioned above, the relationship between leguminous plants and *Bradyrhizobium* bacteria plays a crucial role in the nitrogen cycle and represents one of the most efficient examples of symbiosis in nature. The process of atmospheric nitrogen fixation by these bacteria is essential for the functioning of ecosystems [15]. However, the impact of *Bradyrhizobium* on the availability of other soil elements, such as iron, phosphorus, sulfur, calcium, and manganese, remains the subject of ongoing research. This chapter presents the current state of knowledge on the influence of *Bradyrhizobium* on the content and availability of these elements in the soil and how the bacteria cope with situations of deficiency or excess of these elements in their environment.

### 2.1. Iron

Iron is one of the most important micronutrients for all living organisms, including *Bradyrhizobium*. It plays a crucial role in many cellular processes, including the synthesis of various enzymes—catalase, peroxidase, and cytochrome—and cellular respiration. It is also involved in DNA biosynthesis. Soil usually contains low levels of this element. In an oxygenated environment, iron exists in an oxidized form (iron hydroxide), which makes the particles insoluble [16,17,18,19]. Excessively high or low concentrations of this element are toxic and can alter functioning of *Bradyrhizobium* cells as well as their relationships with plants [16].

#### 2.1.1. Iron Deficiency

Iron deficiency in the soil is a widespread challenge for *Bradyrhizobium*, resulting from the low solubility of iron and the soil’s pH [20]. For this reason, bacteria have developed many mechanisms to acquire and supplement iron deficiencies. One such mechanism is the acquisition of a wide range of siderophores from various microorganisms, enabled by single mutations in the genes of outer membrane receptors, which provide an evolutionary advantage in a changing environment [21,22].

A study by Ong A. [22] on *Bradyrhizobium japonicum* showed that the *fsrB* gene is key to the utilization of various new iron siderophores, as well as synthetic chelators, to meet the bacterium’s iron requirements. Mutants lacking *fsrB* have a defect in iron reduction and its release from siderophores. Iron uptake by *B. japonicum* is highly specific at the stage of binding to outer membrane receptors. However, subsequent stages occur non-specifically, thanks to FsrB, which releases Fe^3^⁺ ions. Iron is then transported to the cytoplasm by the FeoAB transporter, regardless of the type of chelator. The bacterium can effectively utilize a variety of iron sources. Another mechanism of response to low iron levels is to prevent the uptake of this microelement by other microorganisms inhabiting the soil [22].

A study by Dutta K. [20] suggests that low iron levels induce increased expression of the *fadD* gene, which is responsible for deactivating DSF (diffusible signal factor)—signaling molecules produced by other bacteria. This limits their ability to effectively capture iron and reduces their virulence. In this way, *B. japonicum* improves its chances of acquiring iron.

#### 2.1.2. Excess Iron

Excess iron is not common in the environment. However, an excess of this element in bacterial cells results in the formation of high oxidative stress. Iron has the ability to catalyze the formation of reactive oxygen species (ROS) [16,23]. The fight against this challenge is characterized mainly by the use of various exporters, such as P-type ATPases [24,25], the major facilitator superfamily (MFS) [26], or cation diffusion facilitators (CDFs) [27].

Oxidative stress induced by high concentrations of the described element in *Bradyrhizobium* strains with varying symbiotic efficiency triggers the expression of genes encoding antioxidant enzymes, such as superoxide dismutase, glutathione reductase, glutathione synthetase, and bacterioferritin [28]. Bacterioferritin present in *Bradyrhizobium* strains plays a crucial role in iron storage, which may reduce its toxic effects and enhance bacterial survival in iron-rich soils while simultaneously minimizing the risk of oxidative stress [29].

In response to stress caused by elevated iron levels in the soil, *Bradyrhizobium* can increase the activity of nitrogenase, a key enzyme in the nitrogen fixation process. This response allows plants to maintain optimal nitrogen availability despite the presence of stress factors [30]. Additionally, heavy metal-tolerant *Bradyrhizobium* strains facilitate iron uptake by plants, leading to increased chlorophyll synthesis and improved plant nutrition [31,32]. The presence of genes encoding enzymes such as ACC deaminase and auxin suggests that *Bradyrhizobium* may regulate stress hormone levels, particularly ethylene, while also promoting root development, thereby enhancing plant resistance to heavy metals [30,32,33].

### 2.2. Phosphorus

Phosphorus is another essential element for proper functioning of both *Bradyrhizobium* and plant cells [34]. Similar to iron, phosphorus in the soil is present in a poorly available form, which can limit the efficiency of establishing symbiosis and fixing nitrogen by *Bradyrhizobium*. For this reason, phosphorus deficiencies in the soil are often addressed by applying phosphate fertilizers, including rock phosphate [35,36]. Although this solution can be effective in the short term, it is important to note that Earth’s phosphorus reserves are limited, and their depletion is only a matter of time, estimated to be around 200–300 years [37,38]. Therefore, it is crucial to explore more sustainable solutions, for instance, improving the efficiency of phosphorus use in agriculture and optimizing *Bradyrhizobium*’s ability to acquire and utilize phosphorus for enhanced nitrogen fixation [39].

#### 2.2.1. Phosphorus Deficiency

Phosphorus-deficient soils pose a challenge not only for plants but also for symbiotic bacteria. The interaction between leguminous plants and *Bradyrhizobium* under phosphorus-deficient conditions relies on integrated mechanisms that enable optimal utilization of the available phosphorus. Under such conditions, plant roots undergo physiological and metabolic modifications. The plant develops proteoid or cluster roots [40,41], where there is increased synthesis and secretion of organic acids, acid phosphatase [42,43], and flavonoids [44]. These substances increase phosphorus availability in the soil by solubilizing it and other micronutrients in the rhizosphere [45], and they also influence the activity of soil microorganisms [46]. Insufficient phosphorus supply leads to metabolic disturbances in *Bradyrhizobium*, inhibiting processes related to nitrogen fixation and the formation of a large number of root nodules. Under soil acidification conditions with organic acids such as malate or citrate, the activity associated with nitrogen assimilation and amino acid synthesis may be reduced, while the production of organic acids increases [47]. Root nodules are the main phosphorus storage for the plant [48], as a large amount of phosphorus associated with ribosomal RNA is necessary for the process of protein synthesis, including nitrogenase enzymes, which, due to their functions, are particularly susceptible to damage and require constant regeneration [49].

In response to phosphorus deficiency, bradyrhizobial bacteria use a phosphorus transport mechanism in which the pstSCAB operon plays a key role. The genes in this operon encode a transporter with a high affinity for phosphorus, allowing bacteria to effectively absorb phosphorus from the environment, even at very low concentrations of this element. This is particularly important under phosphorus-limited conditions, as the Pst transporter enables efficient acquisition of this element, making it a key mechanism in bacterial adaptation to phosphorus deficiency [50].

#### 2.2.2. Excess Phosphorus

Excess phosphorus in the soil, similar to iron, is not a common occurrence. However, it does not bring any positive benefits for plants or for *Bradyrhizobium*. Excess phosphorus in the nutrient solution caused significant changes in phosphorus metabolism in both the plants and the root nodules of soybean. As a result of the phosphorus excess, the fresh and dry weight of the root nodules decreased by almost 50%, and their number decreased by 35% compared to plants grown in control conditions. Excess phosphorus also affected the total nitrogenase activity, the enzyme responsible for reducing atmospheric nitrogen to ammonia. In plants with excess phosphorus, its activity was reduced by more than half, indicating inhibition of nitrogen fixation processes [51].

### 2.3. Sulfur

Sulfur is an essential element for *Bradyrhizobium*, playing a crucial role in its metabolism. It is a component of amino acids, including cysteine and methionine, which serve as the building blocks of proteins. Additionally, sulfur is a constituent of coenzyme A and other important compounds, contributing to enzymatic and metabolic processes in *Bradyrhizobium* [52,53]. These bacteria also participate in biogeochemical cycles, including the sulfur cycle, which influences their ability to establish symbiosis and fix nitrogen. However, the ability of *Bradyrhizobium* to uptake and utilize sulfur depends not only on its availability in the environment but also on interactions with other microorganisms and the presence of various environmental factors, for instance, heavy metals [54,55].

#### 2.3.1. Sulfur Deficiency

Sulfur deficiency in soybean crops results in delayed and reduced formation of root nodules, which limits the effectiveness of symbiotic nitrogen fixation by *Bradyrhizobium*. Consequently, the number of effective nodules on plant roots decreases, directly impacting seed yield and soybean biomass [56,57]. Sulfur deficiency also reduces nitrogenase activity due to a decrease in the production of amino acids, such as cysteine and methionine, which negatively affects nitrogen processes in plants. Furthermore, sulfur deficiency limits its involvement in critical metabolic reactions, such as protein synthesis, which significantly affects seed quality, particularly the content of sulfur-containing amino acids with high nutritional value [58,59]. Low sulfur concentrations in the soil impair the production of leghemoglobin, glucose, ATP, and ferredoxin in nodules, thereby reducing oxygen availability and limiting the energy required for nitrogen fixation. In the absence of sufficient sulfur, the nitrogen fixation process in plants is severely compromised [60].

Bacteria, including *Bradyrhizobium japonicum*, demonstrate an extraordinary ability to adapt to various environmental conditions, including the utilization of diverse sulfur sources. While sulfur is most commonly supplied as sulfates under laboratory conditions, in natural ecosystems—particularly in soils—most sulfur is present in organic form, associated with various organic compounds. Bacteria have developed specialized metabolic mechanisms that enable them to efficiently utilize these complex sulfur sources, which is crucial for their survival and growth under conditions of limited availability of readily available sulfur [52]. Studies by Sugawara [61] have identified genes in the *Bradyrhizobium japonicum* genome responsible for the utilization of sulfonates, such as bll6451, bll7010, and bll6452, which play a role in converting organic sulfur into a form that can be assimilated by plants. Mutants of *B. japonicum* with disrupted genes involved in sulfur metabolism, such as *ΔssuD*, exhibit reduced metabolic activity and impaired efficiency in symbiotic nitrogen fixation, leading to limited nitrogen assimilation and decreased plant quality [62].

#### 2.3.2. Excess Sulfur

Excessive sulfur application, especially at rates exceeding 40 kg ha^−1^, negatively affects *Bradyrhizobium japonicum*–soybean symbiosis, leading to reduced nodulation efficiency. High sulfur doses limit the availability of sulfates (SO_4_^2−^) in the soil, resulting in lower concentrations of these ions in the root zone, which, in turn, reduces the plant’s ability to absorb sulfur and other essential nutrients [58,63]. Excess sulfur can also lead to the leaching of sulfates from the soil layer occupied by the root system, especially during excessive rainfall, further reducing the availability of SO_4_^2−^ in the soil [63]. As a result, a deficiency of soil sulfates can limit the ability of *Bradyrhizobium japonicum* to establish an effective symbiosis with plants, resulting in a reduced number of effective root nodules and decreased nitrogenase activity [59,60]. The reduced number of nodules leads to lower seed yields and soybean biomass, which is a clear indication of the negative consequences of excess sulfur on symbiotic processes in this system [60].

### 2.4. Calcium

Calcium plays a critical role in the establishment of symbiosis between legumes and bacteria of the genus *Bradyrhizobium*. The initiation of this symbiosis is closely linked to dynamic changes in calcium concentrations within bacterial cells. Flavonoids secreted by the plant induce an increase in cytosolic calcium levels in bacteria [64]. This triggers a signaling cascade that leads to the expression of nod genes. In turn, an increase in calcium concentration in plant cells, prompted by bacterial signals, initiates processes that result in the formation of root nodules [65,66]. An optimal level of calcium in the soil promotes the proper development of root nodules while maintaining the structural integrity of roots, which is essential for effective infection and nodule formation [67].

#### 2.4.1. Calcium Deficiency

Calcium is indispensable for the proper adhesion of bacteria to the root surface. Calcium ions are involved in activating the signaling cascade that leads to the transcription of nod genes, which are responsible for nodule formation [64]. Its deficiency hinders bacterial colonization of roots, the first step in the infection process. Furthermore, calcium is crucial for forming the structural elements of root nodules, enabling their proper development [68].

Calcium plays a key role in stabilizing the hopanoid barrier, which limits oxygen diffusion into the symbiosome, thereby creating conditions conducive to nitrogenase activity. Excess oxygen leads to nitrogenase inactivation. Under calcium-deficient conditions, the hopanoid barrier becomes less stable, resulting in increased oxygen permeability into the symbiosomal space and a significant reduction in nitrogen fixation efficiency [1]. An alternative protective mechanism in the absence of hopanoids is calcium binding to polar membrane lipids, which also shields nitrogenase from oxygen. However, in calcium-deficient conditions, this strategy fails, leading to uncontrolled exposure of nitrogenase to oxygen and further diminishing the efficiency of the symbiotic process [69].

Calcium is also involved in the functionality of leghemoglobin, a protein responsible for binding and regulating oxygen levels in root nodules. A calcium deficiency adversely affects leghemoglobin’s efficiency, exacerbating the challenge of maintaining optimal oxygen conditions in the symbiosome and weakening nitrogenase protection [1,70].

#### 2.4.2. Excess Calcium

High levels of calcium, similar to calcium deficiency, have a significant impact on the symbiotic processes occurring between bradyrhizobial bacteria and leguminous plants, influencing, among other factors, the adhesion of bacteria to root surfaces, their colonization, and nodule formation [67,68]. It is important to note that different bacterial strains may have varying calcium requirements, which determine their ability to establish effective symbiosis [67].

Elevated calcium concentrations, particularly above 12 mM, promote increased bacterial adhesion and enhanced colonization efficiency in most *Bradyrhizobium* strains, indicating their calciphilic nature. These strains exhibit improved attachment to legume roots, facilitating the infection process and nodule formation [68]. However, *Bradyrhizobium japonicum* is an exception, as it is calciphobic and prefers lower calcium concentrations (e.g., 127 µM). At high calcium concentrations, such as 1000 µM, the adhesion of this strain to roots significantly decreases, negatively affecting its ability to colonize effectively and form nodules [67].

Andreev’s study [71] demonstrated that excessive calcium levels may accelerate the aging process of nodule cells. High calcium content was observed in various parts of the cell, such as the symbiosome, vacuole, and apoplast. The accumulation of calcium in these areas led to the disruption of nodule cell structures. It is hypothesized that this phenomenon may occur as a response to the accumulation of abscisic acid (ABA) due to high calcium concentrations in the environment. ABA is a phytohormone responsible for reducing leghemoglobin levels and the rate of atmospheric nitrogen assimilation [72]. During the aging process of nodule cells, studies revealed an increase in calcium concentration in the symbiosomal space. This phenomenon was not observed in immature cells, suggesting that the increase in calcium concentration is associated with the aging process of nodule cells [71].

### 2.5. Manganese

Manganese is the 12th most abundant element on our planet. In the soil, it predominantly occurs in the form of oxides, carbonates, and silicates due to its ability to easily oxidize. As a result of natural erosion processes, significant amounts of manganese are released into the environment, where it becomes integrated into the biogeochemical cycle, meeting the needs of various organisms [73,74]. Within bacterial cells, manganese is primarily recognized for its role in oxidative defense, acting as a cofactor for manganese-dependent superoxide dismutase [75]. Mn^2^⁺ ions participate in redox reactions, generating reactive oxygen species (ROS) that can stimulate root nodule formation and enhance nitrogen fixation efficiency in *Bradyrhizobium* [76]. Its ability to both neutralize and generate reactive oxygen species allows for the maintenance of a delicate redox balance, essential for proper cellular function. Manganese is a crucial element required for the proper functioning of *Bradyrhizobium japonicum* under non-stressful aerobic growth conditions [77]. Unlike other bacteria [78], where manganese is primarily utilized in response to oxidative stress, in *B. japonicum*, this metal plays an essential role in daily metabolic processes [77].

#### 2.5.1. Manganese Deficiency

The study by Hohle and O’Brian [78] demonstrated that the MntH protein, encoded by the *mntH* gene, plays a crucial role in the transport of manganese into the cells of *Bradyrhizobium japonicum*. Mutation of this gene leads to a decrease in manganese levels within the cell, thereby inhibiting bacterial growth under conditions of low manganese availability. The expression of the *mntH* gene is tightly regulated by the Fur (Ferric Uptake Regulator) protein. Under low-manganese conditions, Fur does not bind to the promoter of the *mntH* gene, allowing its expression and enhancing manganese uptake. Interestingly, despite the lack of a functional *mntH* gene, the bacterium is still able to form a symbiotic relationship with the host plant, suggesting the presence of alternative mechanisms for manganese acquisition in symbiotic conditions.

Another study by Hohle and O’Brian [77] indicates that *B. japonicum* exhibits a clear dependence on manganese, which is essential for the activity of glycolytic enzymes, including pyruvate kinase PykM. Unlike other bacteria, this enzyme prefers Mn^2^⁺ over Mg^2^⁺ as a cofactor, highlighting the importance of manganese in the metabolism of this microorganism. The bacterium possesses a specialized manganese transport system, including a high-affinity Mn^2^⁺ transporter and the membrane channel MnoP, which allows efficient manganese uptake even in environments with very low manganese concentrations. *B. japonicum* cannot compensate for manganese deficiency using other enzyme-independent mechanisms, emphasizing the critical role of manganese in the basic cellular processes of this organism.

Both studies demonstrated that manganese plays a key, multifunctional role in the metabolism of *B. japonicum*, being essential for both growth and the proper functioning of glycolytic enzymes as well as antioxidative mechanisms [75,78].

#### 2.5.2. Excess Manganese

Excessive manganese concentrations in soil and in the growth environment of plants in symbiosis with root nodule bacteria, such as *Bradyrhizobium japonicum*, can exert a negative impact on the biological processes of these microorganisms. In low-pH soils, where aluminum and manganese ion concentrations are elevated, the symbiosis between leguminous plants and root nodule bacteria may be significantly restricted [79]. Specifically, manganese, similar to copper, affects the activity of nodule bacteria by disrupting the formation of new nodules and inhibiting bacterial function in already established nodules [80,81].

Furthermore, under conditions of high manganese concentrations, *B. japonicum* exhibits reduced metabolic activity, negatively affecting its ability to carry out essential cellular processes, including energy metabolism. Fertilizer studies have shown that even small excesses of manganese, particularly in combination with other metals, can have a detrimental effect on bacterial viability, emphasizing the importance of optimal manganese concentrations for the proper functioning of these microorganisms [82,83,84].

The influence of selected elements on *Bradyrhizobium* is shown in Table 1.

## 3. Adaptation of Bradyrhizobia to Environments Rich in Heavy Metals and Radionuclides

Some heavy metals, such as lead, mercury, and arsenic, as well as radionuclides, pose a serious threat to microorganisms, including bacteria from the *Bradyrhizobium* genus [85,86,87,88]. Exposure to these elements can disrupt the metabolism and overall functioning of these bacteria [85]. In response to such conditions, microorganisms can develop different defense strategies [89]. Research on the impact of these components on the evolution and defense mechanisms of *Bradyrhizobium* is crucial for improving crop productivity.

### 3.1. Lead

Lead is one of the metals that negatively affects *Bradyrhizobium* bacteria. However, some strains show tolerance to high concentrations of lead, which may support bioremediation processes. Studies describe three *Bradyrhizobium* strains isolated from root nodules that can tolerate lead acetate concentrations of up to 300 μg mL^−1^. These strains were also capable of forming functional nodules on the roots of Retama monosperma in the presence of lead [85]. This indicates their ability to maintain symbiosis even in contaminated environments. Genome sequencing studies revealed that these strains are closely related to the species *Bradyrhizobium algeriense*, which also tolerates heavy metals [90].

The molecular mechanisms responsible for lead tolerance in *Bradyrhizobium* include factors like efflux pumps and membrane transporters, which help remove toxic metal ions from bacterial cells [85]. Additionally, genes such as *nodC*, which are important for nodule formation, may play a role in regulating stress responses caused by lead exposure. The ability to tolerate lead may also be linked to maintaining symbiotic functions. These mechanisms include enzymes that neutralize the toxic effects of metals and structural changes in the cell membrane that help the bacteria adapt to polluted environments [85].

### 3.2. Mercury

Mercury is one of the most toxic heavy metals, posing a threat to human, animal, and plant health, as well as causing environmental degradation. Some *Bradyrhizobium* strains have shown tolerance to high mercury concentrations, which could be useful in mercury-contaminated environments [85]. Other studies indicate that certain *Bradyrhizobium* strains exhibit mercury reductase activity (gene *merA*), which converts Hg^2^⁺ into the less toxic Hg⁰ [89]. Strains such as *Bradyrhizobium canariense*, which exhibit higher *merA* activity, may contribute to better soil detoxification [89].

Other studies on *Bradyrhizobium* isolated from mercury mines also showed high tolerance to this metal, with a minimum inhibitory concentration (MIC) above 150 μM [91]. Additionally, mercury-tolerant strains were also resistant to other heavy metals, such as zinc, lead, and chromium, which may indicate the presence of multi-component detoxification systems [91]. Strains belonging to clade III were sensitive to mercury, suggesting that tolerance mechanisms may depend on phylogenetic affiliation and the environmental conditions in which the bacteria evolved [91].

### 3.3. Arsenic

Arsenic in the soil affects its physicochemical and enzymatic properties, and its bioavailability depends on speciation and factors such as organic matter, pH, and the presence of calcium [92]. This element has a negative impact on *Bradyrhizobium*, disrupting its biochemical activities in plant roots. In the presence of arsenic, bacteria produce more siderophores, which suggests an attempt to bind and neutralize the stress factor, while also increasing auxin production [93]. The produced siderophores play a role as chelating compounds, reducing the toxicity of the element for both bacteria and plants. The increase in the production of these important compounds is especially noticeable in plants inoculated with *Bradyrhizobium*, indicating the activation of defense mechanisms under stress conditions [94]. Additionally, arsenic affects the activity of antioxidant enzymes, such as peroxidases, including ascorbate peroxidase (APx), which, when *Bradyrhizobium* is inoculated, are used to neutralize free radicals produced due to oxidative stress caused by the metal [94]. According to studies, arsenic contributes to changes in the metabolism of microorganisms, affecting their ability to form nodules, which is a result of the toxic effect of the described element on *Bradyrhizobium*, disrupting the symbiotic processes with plants [93]. Moreover, different *Bradyrhizobium* strains exhibit varying tolerance to arsenic, which affects the efficiency of symbiosis and adaptation to contaminated environments [95].

### 3.4. Radionuclides

Radionuclides, or radioactive isotopes, also have a significant impact on bacteria from the *Bradyrhizobium* genus. Studies conducted on soils contaminated with radionuclides, antibiotics, and heavy metals have shown that these microorganisms possess diverse genes related to adaptation to toxic environments, suggesting that the presence of radionuclides (e.g., ^3^H, ^137^Cs, ^90^Sr, americium-241, cerium-244 and plutonium-239, 240) promotes the selection of resistant strains [88]. Metagenomic studies revealed that isolates found in contaminated soils often have genes coding for ABC transport proteins and two-component regulators, which help remove toxic ions such as nickel, cobalt, and zinc [88]. Additionally, in *Bradyrhizobium* strains, increased expression of genes related to arsenate detoxification and metal stress regulation, such as *arsC* and *czcR*, was detected, which may enhance their survival in environments contaminated with radionuclides [88].

Similar relationships have also been observed in other environments with high levels of heavy metals and antibiotics. Previous studies have shown that exposure to metals and radionuclides leads to the co-selection of genes resistant to both antibiotics and metals, which may serve as an adaptive mechanism for *Bradyrhizobium* in contaminated habitats [96,97].

## 4. Bradyhizobia and Pesticides Used in Agriculture

Pesticides are widely used in agriculture to protect crops from harmful factors like weeds and pests because of their high effectiveness and quick action [98]. However, there is increasing concern about the potential harmful effects of these substances on naturally occurring microorganisms in the soil, especially *Bradyrhizobium* species [99,100]. The use of pesticides in legume cultivation may disturb the balance of the rhizosphere microbiome and negatively affect symbiosis efficiency [101].

### 4.1. Herbicides and Their Effect on Bradyrhizobia

Herbicides, which are a type of pesticide, are widely used in agriculture, including in the cultivation of plants from the Fabaceae family, to control weeds. Their presence in the soil can affect bacteria from the *Bradyrhizobium* genus. Studies highlight that herbicides may reduce the population of these bacteria in the soil, disrupting their ability to form effective symbiosis with the host, which affects the nitrogen fixation process [102]. Furthermore, some herbicides can induce oxidative stress in bacteria, leading to changes in the expression of genes related to stress response and detoxification [103]. As a result, prolonged use of herbicides may promote the selection of strains resistant to these substances, which alters the composition of the soil microbiota [104]. This is a relevant issue in the context of sustainable agriculture.

#### 4.1.1. Glyphosate

One of the commonly used herbicides is glyphosate (GP), which has a broad spectrum of activity against both perennial and annual weeds [105]. Its negative impact on the soil microbiome involves inhibiting the shikimate pathway, which reduces the availability of important nutrients [106]. The most effective way to defend against the harmful effects of glyphosate is its degradation to aminomethylphosphonic acid (AMPA), which is also toxic, but to a lesser extent [107,108]. Some *Bradyrhizobium* species possess the glycine oxidase gene (*thiO*), which participates in the oxidation of GP to AMPA, thereby influencing the concentration of the herbicide in the soil (Figure 1) [109]. These studies suggest that these strains have a greater ability to survive in environments contaminated with these harmful chemicals, which helps maintain agricultural productivity.

#### 4.1.2. Sulfentrazone

Sulfentrazone is an herbicide applied before the germination process and is known for its mobility in certain types of soil [110]. Its presence in the environment promotes the growth of fungi and actinomycetes, while inhibiting the growth of bacteria [111].

Recent studies show that the symbiosis between *Bradyrhizobium* sp. and plants may enhance tolerance to sulfentrazone, as evidenced by the reduction in residual herbicide concentrations in the soil. This phenomenon may represent a bacterial adaptation mechanism to the environment [112]. Plants inoculated with the bacteria also showed an improvement in the fresh shoot mass compared to non-inoculated plants, especially at higher herbicide doses [112]. The increased C-CO_2_ production in soils containing *Bradyrhizobium* suggests the activation of metabolic mechanisms related to sulfentrazone detoxification [112]. It can also be assumed that these bacteria improve the microbiological activity of the soil, which leads to better degradation of the herbicide [112].

#### 4.1.3. Chlorophenoxy Herbicides

Chlorophenoxy herbicides are used to control broadleaf weeds. Some microorganisms can degrade certain pesticides in this group, such as 2,4-dichlorophenoxyacetic acid (2,4-D), indicating the potential use of microorganisms in bioremediation [113,114]. *Bradyrhizobium* sp. RD5-C2 has the ability to degrade chlorophenoxyacetic acids due to the presence of two distinct cad gene clusters [115].

Phylogenetic analysis suggests that this bacterium, initially capable of breaking down other compounds than 2,4-D, acquired the cad1 cluster through horizontal gene transfer, allowing it to effectively degrade 2,4-D [115]. This ability, along with the widespread presence of *Bradyrhizobium* in the soil, can contribute to improving soil and water quality and supporting ecological balance in agricultural areas affected by chlorophenoxy herbicide contamination.

#### 4.1.4. Flumioxazin and Imidazolinone

Flumioxazin is a pre-emergence herbicide with a broad spectrum of activity [116]. Imidazolinone herbicides, such as imazapyr, are used mainly to control grasses and weeds [117]. Both herbicides are commonly used in agriculture as replacements for glyphosate and can have a short-term, positive effect on rhizobial bacteria [118]. A temporary, positive effect on nitrogen fixation by these microorganisms was also observed, indicating improved efficiency in nitrogen-related processes while maintaining soil microbiome balance [118]. It can be assumed that *Bradyrhizobium* benefits from changes in nutrient availability or ecological niche caused by these applied herbicides, allowing them to colonize the soil environment more effectively.

#### 4.1.5. Atrazine

*Bradyrhizobium* plays a role in atrazine degradation in the soil, as described in studies on processes in the rhizosphere of plants [119,120,121]. Higher numbers of *Bradyrhizobium* were found in the rhizosphere of vetiver plants (*Chrysopogon zizanioides* L.) compared to soil without these plants, suggesting that *Bradyrhizobium* may be key in the degradation of atrazine [119]. Along with other bacteria that degrade this herbicide, such as *Arthrobacter, Nocardioides*, and *Rhodococcus*, these microorganisms form a network that effectively contributes to the removal of atrazine from the soil [119]. Additionally, studies on the metabolism of atrazine showed that these plants can metabolize the herbicide, creating various metabolites in shoots and roots [119]. These results suggest that microorganisms like *Bradyrhizobium* can use these metabolites, further contributing to atrazine degradation and improving soil properties by eliminating toxic substances. In studies on soybeans exposed to atrazine, the introduction of the *Bradyrhizobium* AC20 strain reduced the herbicide’s toxicity [120]. This suggests that changes in the microbiological composition of the soil, influenced by atrazine-degrading bacteria, could impact the plant’s ability to adapt in the presence of the herbicide [120].

Studies on plants such as *Liriope striata* and *Carya ferrea* also confirm that *Bradyrhizobium* is one of the dominant microorganisms in soil where atrazine has been applied [121]. Genes responsible for atrazine degradation, such as *atzD, atzE*, and *atzF*, were detected among the microorganisms, indicating that bacteria like *Bradyrhizobium* can potentially accelerate the degradation of the pesticide and reduce its toxic effects on plants [121]. However, studies also suggest the presence of other atrazine-degrading microorganisms, such as *Agrobacterium rhizogenes, Candidatus muproteobacteria*, and *Micromonospora*, pointing to the broad role of the soil microbiome in this process [121].

### 4.2. Effects of Insecticides and Fungicides on Bradyrhizobium

*Bradyrhizobium* shows some potential for adapting to the presence of insecticides and fungicides, but research indicates that these substances negatively affect bacterial survival [99]. In the study, strains of *Bradyrhizobium japonicum* SEMIA 5079, *Bradyrhizobium elkanii* SEMIA 587, and *Bradyrhizobium diazoeficiens* SEMIA 5080 were exposed to pesticides containing pyraclostrobin, methyl thiophanate, and fipronil, which led to changes in the morphology of these strains (smaller colony size), though these changes were reversible once the stressor was removed [99]. This adaptive potential suggests that under suitable conditions, the negative effects of pesticides can be reduced, improving symbiosis efficiency in soils exposed to these chemicals.

### 4.3. Bradyrhizobia as Biopesticides

Microorganisms belonging to *Bradyrhizobium* have potential as biopesticides [122]. *Macrophomina phaseolina* is a pathogenic fungus that causes root rot in soybean [123]. Recent studies describe the effect of *Bradyrhizobium japonicum* 406 in reducing the occurrence and severity of this disease [122]. The mechanism is based on strengthening plant resistance by activating defense enzymes (peroxidase, polyphenol oxidase, catalase) and synthesizing specific chemical compounds, such as flavonoids and isoflavones, which have antimicrobial properties [122]. Because of these properties, it can be an effective alternative to traditional pesticides.

The classification of pesticides and their effects on *Bradyrhizobium* is provided in Table 2.

## 5. Conclusions

This review describes the effect of selected elements, heavy metals and pesticides on one of the most important groups of rhizobia bacteria, the genus *Bradyrhizobium*, which not only shapes many processes occurring in the natural soil environment but is also widely used as a biofertilizer, especially in field crops of legume plants. This study shows that bradyrhizobia have developed a number of adaptations to diverse soil conditions, in which there are excesses or deficiencies of certain compounds or which are contaminated with mutagenic factors. Moreover, they are not only able to survive unfavorable environmental conditions, but they have also developed pathways that allow them to function properly under these stressful conditions, resulting in a better adaptation of the soil environment for other bacteria and plants occurring there. We believe that understanding the impact of various factors affecting bradyrhizobia will help in the selection of strains, which in the future can be used as very effective bacterial vaccines supporting plant growth and helping to eliminate the harmful effects of human activity on the soil environment.

## Figures and Tables

**Figure 1 cimb-47-00205-f001:**
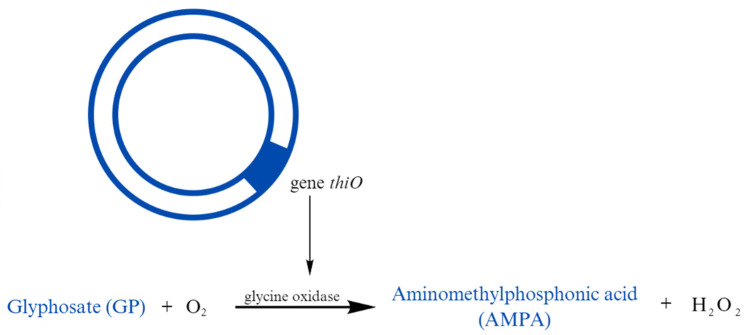
Reaction scheme for the conversion of GP to AMPA.

**Table 1 cimb-47-00205-t001:** Classification of selected elements and their effects on *Bradyrhizobium*, as described in the literature.

Type of Element	Studied Bacterial Strain	Effect of Elements	References
Iron	*Bradyrhizobium japonicum*	Under iron deficiency, the signaling molecule DSFs from other bacteria are deactivated	Dutta K. et al. (2023) [20]
	*Bradyrhizobium japonicum*	Under iron deficiency, *fsrB* expression is upregulated	Ong A., O’Brian M.R. (2023) [22]
Phosphorus	*Bradyrhizobium* sp.	Under phosphorus deficiency, reduction in nitrogen assimilation	Williams A. et al. (2022) [47]
Sulfur	*Bradyrhizobium* sp.	Under sulfur deficiency, reduction in metabolic activity	Habetamu G. et al. (2021) [59]
Calcium	*Bradyrhizobium* spp.	Under calcium deficiency, inactivation of nitrogenase	Ledermann R. et al. (2021) [1]
Manganese	*Bradyrhizobium japonicum*	Manganese excess limits symbiosis	Pisarek I. (2023) [79]
	*Bradyrhizobium japonicum*	Limitation of cellular processes	Brzezińska A., Mrozek-Niećko A. (2021) [84]

**Table 2 cimb-47-00205-t002:** Classification of pesticides and their effects on *Bradyrhizobium* as described in the literature.

Type of Pesticide	Pesticide	Soil Persistence	Studied Bacterial Strain	Effect of Pesticide	References
Herbicides	Glyphosate	Low to medium persistence	*Bradyrhizobium* sp.	Bacteria have a *thiO* gene, which oxidizes GP to AMPA	Hernánde Guliaro K. et al. (2021) [109]
	Sulfentrazone	Medium persistence	*Bradyrhizobium japonicum* SEMIA 5079	Increasing herbicide tolerance in plants	Mielke K.C. et al. (2020) [112]
	2,4-dichlorophenoxyacetic acid	Low to medium persistence	*Bradyrhizobium* sp.	The bacterium has a *cad1* cluster	Hayashi S. et al. (2020) [115]
	Flumioxazin	Medium persistence	*Bradyrhizobium* sp.	Temporary positive effect on nitrogen fixation	Araújo A. S. et al. (2023) [118]
	Imazapyr	High persistence			
	Atrazine	Medium to high persistence	*Bradyrhizobium* sp.	Potential impact on pesticide degradation	Zhang, F. et al. (2023) [119]
			*Bradyrhizobium japonicum* AC20	Mitigation of pesticide effect on soil	Jiang, D et al. (2023) [120]
			*Bradyrhizobium* sp.	Possess genes degrading atrazine (*atzD, atzE, atzF*)	Aguiar, L et al. (2020) [121]
Insecticides and fungicides	Pyraclostrobin	Medium persistence	*Bradyrhizobium* *japonicum* SEMIA 5079, *Bradyrhizobium elkanii* SEMIA 587, *Bradyrhizobium diazoeficiens* SEMIA 5080	Morphological changes in strains	Rodrigues T. et al. (2020) [99]
	Thiofanate-methyl	Medium to high persistence			
	Fipronil	High persistence			

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
