# Peer review of "Adaptations of the Genus Bradyrhizobium to Selected Elements, Heavy Metals and Pesticides Present in the Soil Environment"

_cimb, 2025, doi:10.3390/cimb47030205_

Round 1
Reviewer 1 Report
Comments and Suggestions for Authors
Overall, this review is pretty interesting. I suggest that authors may add one illustration for each section such as for this section, an illustration summarize the adaptation of bradyrhizobia to low and high concentrations of elements such as iron, phosphorus, sulfur, calcium and manganese can be attractive. The English writing also should be improved by native speakers.
Comments on the Quality of English LanguageGood, but English writing also should be improved by native speakers.
Author Response
Comments 1: Overall, this review is pretty interesting. I suggest that authors may add one illustration for each section such as for this section, an illustration summarize the adaptation of bradyrhizobia to low and high concentrations of elements such as iron, phosphorus, sulfur, calcium and manganese can be attractive. The English writing also should be improved by native speakers.
Comments on the Quality of English Language: Good, but English writing also should be improved by native speakers.
Response 1: We would like to thank you very much for your review of our article and for your valuable comments. We decided not to include a graphic summarizing the adaptation of bradyrhizobia to low and high concentrations of the given elements, because we believe that it was better to present the results in a table, where there are also references to specific literature on the cited studies. We used a similar approach in the case of the effect of pesticides on bradyrhizobia. The original publications contain appropriate figures/diagrams showing the described phenomena, we did not want to duplicate the content in this case. Only in subsection 4.1.1. we added a graphic showing the effect of the glycine oxidase gene on the oxidation of GP to AMPA, which was missing in the original sources. We hope that the presentation of these issues in the form of tables is as attractive as if graphics were added. The manuscript was also double-checked for correct use of English. We hope that this has made the work clearer and more linguistically correct.
Reviewer 2 Report
Comments and Suggestions for Authors
I don't understant why the article is mainly referred to soybean. Are there any other examples missing? If these sudies refer to soybean, why not mention it in the title? Being a review there is only a suggestion to improve the knowledge on this topic, what about the other plant species?
In some cases there is only one article to explain your point of view, why did you write a review? It seems there are few studies, tests must to be reproduce to understand if it is correct the results. Did you miss some references?
line 19= in the presence of heavy metals such as ................, as well as
line 36= two spaces between phytohormone and
line 168= are the changes in metabolism only for soybean? In this case not so important, if the execess is also referred to further plants why did you write soybean?
line 184= Sulfur deficiency in soybean. Is this going to happen only in soybean?
line 212= the sulfur execess is referred to soybean, so why didn't you mention in the title, not in the abastract or introduction?
line 220-222= only for soybean plants? I don't understand why talking only about soybean and not ohter plant species.
line 377= instead of arsen , maybe you wanted to write arsenic
line 386-387= are both peroxidase with a different acronyms, APx and Px? It sounds strange. There are different peroxidases, could you check it , please?
Author Response
We would like to thank you very much for your review of our article and for your valuable comments. Below we have included responses to individual comments. We hope that the revised manuscript will be clearer and will add scientific value to the current state of knowledge about bradyrhizobia.
Comments 1: I don't understant why the article is mainly referred to soybean. Are there any other examples missing? If these sudies refer to soybean, why not mention it in the title? Being a review there is only a suggestion to improve the knowledge on this topic, what about the other plant species?
In some cases there is only one article to explain your point of view, why did you write a review? It seems there are few studies, tests must to be reproduce to understand if it is correct the results. Did you miss some references?
Response 1: We wanted to devote our work to a relatively small group of rhizobial bacteria belonging only to the genus Bradyrhizobium. Certainly, if we described all bacteria that fix molecular nitrogen in symbiosis with Fabaceae plants, there would be many more of these original studies. The genus Bradyrhizobium has recently gained importance, so we wanted to present as many studies on it as possible. Otherwise, it would "get lost" among other genera described in original and review articles, especially in the context of a large pool of articles devoted to another genus - Rhizobium.
Most studies on bradyrhizobia are actually based on their interactions with soybean. It is a model plant for such analyses, with enormous agricultural and industrial potential. That is why there are so many studies on the relationship between bradyrhizobia and soybean. But we did not want to limit ourselves to soybean. If we only found studies on other legumes, we wanted to include them in this work as well. So this article is not just about soybean. However, we certainly agree with the comment that most of the studies cited here concern symbiosis with soybean. Furthermore, we have tried to keep the studies described here as up-to-date as possible, hence their limited number.
Comments 2: line 19= in the presence of heavy metals such as ................, as well as
Response 2: We have made the appropriate correction to the text.
Comments 3: line 36= two spaces between phytohormone and
Response 3: We have made the appropriate correction to the text.
Comments 4: line 168= are the changes in metabolism only for soybean? In this case not so important, if the execess is also referred to further plants why did you write soybean?
Response 4: The metabolic changes discussed in the study are believed that are not exclusive to soybeans; however, soybeans were chosen as the model plant due to their high efficiency in symbiotic interactions with Bradyrhizobium bacteria. The findings may have broader implications for other leguminous plants, but the study specifically focuses on soybeans to provide a detailed and controlled analysis of the mechanisms involved. As for phosphorus excess, here, to our knowledge, studies in this respect of bradyrhizobia-mediated symbiosis have only been conducted in soybean model plants.
Comments 5: line 184= Sulfur deficiency in soybean. Is this going to happen only in soybean?
Response 5: Similar to above - soybeans serve as a model plant and are one of the most extensively studied species regarding symbiosis with Bradyrhizobium bacteria. Due to their well-characterized symbiotic interactions and high efficiency in nitrogen fixation, soybeans provide a valuable system for investigating the impact of sulfur deficiency on legume-bacteria symbiosis. While similar deficiencies may occur in other leguminous plants, soybeans offer the most comprehensive dataset for analyzing these processes, making them the preferred model for such studies. Thus, the original research publications that describe bradyrhizobia provide us with data primarily for soybean.
Comments 6: line 212= the sulfur execess is referred to soybean, so why didn't you mention in the title, not in the abastract or introduction?
Response 6: We wanted to maintain a uniform structure of the work. Although most studies on the genus Bradyrhizobium concern soybean, we did not want to exclude those describing bradyrhizobia in symbiosis with other plants. There are not so many such studies if we consider only the genus Bradyrhizobium, and not other rhizobial bacteria. However, we believe that this topic devoted exclusively to this genus is needed in the literature.
Comments 7: line 220-222= only for soybean plants? I don't understand why talking only about soybean and not ohter plant species.
Response 7: The studies we found in the topic described here concerned the association of Bradyrhziobium bacteria only with model soybean plants. There are still few original research papers on such studies of bradyrhizobia in symbiosis with plants. And we wanted to outline the existence of such a topic, , although so far in the literature it refers to soybean.
Comments 8: line 377= instead of arsen , maybe you wanted to write arsenic
Response 8: Yes, sorry for the mistake, it was about arsenic. We have made the appropriate correction to the text.
Comments 9: line 386-387= are both peroxidase with a different acronyms, APx and Px? It sounds strange. There are different peroxidases, could you check it , please?
Response 9: Thank you for this comment, the study concerned the analysis of total peroxidase (Px) and ascorbate peroxidase (APx). We have made the appropriate correction to the text.
Reviewer 3 Report
Comments and Suggestions for Authors
This paper presents a study on the behavior of bacteria of the genus Bradyrhziobium in soil with high concentrations of various elements, heavy metals, radionuclides, and pesticides. Although it is mentioned that these bacteria also have the role of “to degrade such harmful compounds”, this aspect is rather poorly described in the text.
Comments (in order of the text)
- Line 45 “reducing heavy metal contaminations” – the presence and multiplication of Bradyrhziobium bacteria in soil cannot reduce heavy metal contaminations. If there are studies regarding this, they should be shown.
- 318 – excess
- In some paragraphs of the text, there are quite a few descriptions of the behavior of other groups of microorganisms under different environmental conditions in the soil. These paragraphs should be reduced and presented especially those related to bacteria of the genus Bradyrhziobium.
For example, In table 1 – instead of “Environmental strains“ in general, aspects related to the studied bacteria should be shown.
- Species and genera of microorganisms and plants should be written in italics, e.g. lines 350, 490
- 387 “antioxidant enzymes, such as peroxidase (APx)“ – ascorbate peroxidase (Apx)
- In section 3.4 Radionuclides – the authors describe the effect of “toxic ions” on bacteria; no radionuclide is mentioned. This should be checked, completed and explained.
- Lines 406-410 – same comment, what are those radionuclides?
- 445 – “leads to better tolerance to sulfentrazone, as evidenced by the reduction of residual herbicide concentrations in the soil” – what is the relationship between tolerance to the pesticide and the reduction of pesticide concentration in the soil? This should be explained.
- - 459 - 2,4-D - ?
- As in table 1, table 2 also uses the expression “Environmental strains”. What strains? Are they strains of species from the genus Bradyrhizobium? It should be explained.
The entire text should show more clearly the effect of various substances on bacteria of the genus Bradyrhizobium, and explain in more detail, if known, the aspects related to the mechanisms of adaptation of the metabolism of these bacteria to the compounds in the environment. The content of the text should be consistent with the title.
Author Response
We would like to thank you very much for your review of our article and for your valuable comments. Below we have included responses to individual comments. We hope that the revised manuscript will be clearer and will add scientific value to the current state of knowledge about bradyrhizobia.
This paper presents a study on the behavior of bacteria of the genus Bradyrhziobium in soil with high concentrations of various elements, heavy metals, radionuclides, and pesticides. Although it is mentioned that these bacteria also have the role of “to degrade such harmful compounds”, this aspect is rather poorly described in the text.
Comments (in order of the text)
Comments 1: Line 45 “reducing heavy metal contaminations” – the presence and multiplication of Bradyrhziobium bacteria in soil cannot reduce heavy metal contaminations. If there are studies regarding this, they should be shown.
Response 1: Thank you very much for this comment. Indeed, in the work we describe studies on the adaptation of bradyrhizobia and their plant symbionts to heavy metals and such a formulation may be too far-reaching conclusion. We have made the appropriate correction to the text.
Comments 2: 318 – excess
Response 2: We have made the appropriate correction to the text.
Comments 3: In some paragraphs of the text, there are quite a few descriptions of the behavior of other groups of microorganisms under different environmental conditions in the soil. These paragraphs should be reduced and presented especially those related to bacteria of the genus Bradyrhziobium.
For example, In table 1 – instead of “Environmental strains“ in general, aspects related to the studied bacteria should be shown.
Response 3: We agree that we sometimes used too broad formulations, especially when the studies concerned a larger group of bacteria. Now, where bacteria of the genus Bradyrhizobium were concerned, we tried to clearly emphasize this. In Table 1, instead of "environmental strains", we also wrote that it also concerned bradyrhizobial strains, even if the species was not specified in these studies. We hope that this will now be clearer.
Comments 4: Species and genera of microorganisms and plants should be written in italics, e.g. lines 350, 490
Response 4: The lack of use of italics was due to an oversight on our part. We have made appropriate corrections in the text.
Comments 5: 387 “antioxidant enzymes, such as peroxidase (APx)“ – ascorbate peroxidase (Apx)
Response 5: Thank you for this comment, the study concerned the analysis of total peroxidase (Px) and ascorbate peroxidase (APx). We have made the appropriate correction to the text.
Comments 6: In section 3.4 Radionuclides – the authors describe the effect of “toxic ions” on bacteria; no radionuclide is mentioned. This should be checked, completed and explained.
Response 6: We have supplemented the examples of radionuclides in the text.
Comments 7: Lines 406-410 – same comment, what are those radionuclides?
Response 7: We have made the appropriate correction to the text.
Comments 8: 445 – “leads to better tolerance to sulfentrazone, as evidenced by the reduction of residual herbicide concentrations in the soil” – what is the relationship between tolerance to the pesticide and the reduction of pesticide concentration in the soil? This should be explained.
Response 8: We have made the appropriate correction to the text.
Comments 9: 459 - 2,4-D - ?
Response 9: 2,4-D is an abbreviation for 2,4-dichlorophenoxyacetic acid - this abbreviation was already introduced in the previous paragraph.
Comments 10: As in table 1, table 2 also uses the expression “Environmental strains”. What strains? Are they strains of species from the genus Bradyrhizobium? It should be explained.
Response 10: Of course, as before, we have specified these strains in the table 2 as bradyrhizobia.
Comments 11: The entire text should show more clearly the effect of various substances on bacteria of the genus Bradyrhizobium, and explain in more detail, if known, the aspects related to the mechanisms of adaptation of the metabolism of these bacteria to the compounds in the environment. The content of the text should be consistent with the title.
Response 11: We tried to describe as accurately as possible all the original studies that we could find on bradyrhizobia and their adaptations to selected environmental factors. We wanted to devote our work to a relatively small group of rhizobial bacteria belonging only to the genus Bradyrhizobium. Certainly, if we described all bacteria that fix molecular nitrogen in symbiosis with Fabaceae plants, there would be many more of these original studies. The genus Bradyrhizobium has recently gained importance, so we wanted to present as many studies on it as possible. Otherwise, it would "get lost" among other genera described in original and review articles, especially in the context of a large pool of articles devoted to another genus - Rhizobium. Furthermore, we have tried to keep the studies described here as up-to-date as possible, hence their limited number. There are not many studies on the adaptation of the Bradyrhizobium genus alone, but due to its great importance, we wanted to present these selected aspects as best as possible, which were sometimes described very generally in the original papers. Despite everything, we hope that the mechanisms described by us in this review paper can help in further studies of this rhizobial genus.